# Facing the Challenge to Mimic Breast Cancer Heterogeneity: Established and Emerging Experimental Preclinical Models Integrated with Omics Technologies

**DOI:** 10.3390/ijms26104572

**Published:** 2025-05-10

**Authors:** Alessia Ciringione, Federica Rizzi

**Affiliations:** 1Laboratory of Biochemistry, Molecular Biology and Oncometabolism, Department of Medicine and Surgery, University of Parma, Via Volturno 39, 43125 Parma, Italy; alessia.ciringione@unipr.it; 2National Institute of Biostructure and Biosystems (INBB), 00165 Rome, Italy

**Keywords:** breast cancer cell lines, PDOs, organ-on-chip technologies, GEMMs, xenograft mouse models, tumorheterogeneity, omics technologies

## Abstract

Breast cancer (BC) is among the most common neoplasms globally and is the leading cause of cancer-related mortality in women. Despite significant advancements in prevention, early diagnosis, and treatment strategies made over the past two decades, breast cancer continues to pose a significant global health challenge. One of the major obstacles in the clinical management of breast cancer patients is the high intertumoral and intratumoral heterogeneity that influences disease progression and therapeutic outcomes. The inability of preclinical experimental models to replicate this diversity has hindered the comprehensive understanding of BC pathogenesis and the development of new therapeutic strategies. An ideal experimental model must recapitulate every aspect of human BC to maintain the highest predictive validity. Therefore, a thorough understanding of each model’s inherent characteristics and limitations is essential to bridging the gap between basic research and translational medicine. In this context, omics technologies serve as powerful tools for establishing comparisons between experimental models and human tumors, which may help address BC heterogeneity and vulnerabilities. This review examines the BC models currently used in preclinical research, including cell lines, patient-derived organoids (PDOs), organ-on-chip technologies, carcinogen-induced mouse models, genetically engineered mouse models (GEMMs), and xenograft mouse models. We emphasize the advantages and disadvantages of each model and outline the most important applications of omics techniques to aid researchers in selecting the most relevant model to address their specific research questions.

## 1. Introduction

Breast cancer (BC) is the leading cause of cancer-related mortality among women, with an estimated incidence of 47% in 2022 [1]. Despite substantial improvements in the clinical management of BC over the past two decades, predicting the actual behavior of breast tumor pathophysiology remains challenging. The most common and widely accepted histopathological classification of BC is based on the expression of cell surface receptors: the estrogen receptor (ER), the progesterone receptor (PR), and the human epidermal growth factor receptor 2 (HER2). The luminal A and luminal B subtypes are characterized by the presence of the ER, but they differ in the expression levels of the PR and Ki-67 [2]. Indeed, while both can benefit from endocrine therapy and chemotherapy, luminal B tumors exhibit lower 5- and 10-year survival rates compared to luminal A [3]. According to the American Society of Clinical Oncology—College of American Pathologists (ASCO–CAP) guidelines, the HER2-enriched subtype includes tumors that display HER2 positivity in ≥10% of tumor cells or exhibit HER2 gene amplification [4]. Patients in this subgroup can be treated with monoclonal antibodies targeting the HER2/NEU protein, in addition to surgery and chemotherapy [5]. The basal-like subtype, also known as triple-negative breast cancer (TNBC), is characterized by a low or absent expression of ER, PR, and HER2 [6]. As a result, TNBC does not respond to endocrine therapy or drugs targeting HER2, leading to the worst prognosis [7].

In addition to the well-established intertumoral heterogeneity that reflects the differences among patients, there is also a significant diversity in tumor cell subpopulations within a patient’s primary tumor or metastasis, known as intratumoral heterogeneity. Microarray-based gene expression profiling studies and extensive exome/genome sequencing have enhanced our understanding of the remarkable heterogeneous landscape of BC [6,8,9], suggesting the potential to use molecular data to predict responses to chemotherapy and to guide the development of personalized therapeutic strategies [10]. More recently, single-cell DNA and RNA sequencing have demonstrated that long-term drug treatment drives the phenotypic evolution of cancer cells into drug-resistant subtypes primarily through the selection of pre-existing resistant clones that dominate after treatment [11]. Therefore, unraveling and targeting the complex heterogeneity of BC at various levels is the focus of current and future research. In this context, selecting the most appropriate model for preclinical research is critical to adequately study the genetic and non-genetic variability of this tumor, providing novel insights into overcoming therapeutic resistance and achieving personalized treatments for each patient.

In this review article, we describe the main BC models currently used in preclinical research, including cell lines, patient-derived organoids, organ-on-chip technologies, carcinogen-induced mouse models, genetically engineered mouse models, xenografts, and humanized mouse models (Figure 1). We outline the most important applications and research areas where all models are currently employed. Importantly, we highlight the contribution of powerful omics techniques that have enabled the molecular characterization of well-established in vitro and in vivo models and generated data to evaluate how these models succeed or fail in recapitulating the heterogeneous landscape of human BC. We discuss the advantages, disadvantages, and limitations of each model to provide investigators with an updated overview that will assist them in choosing the most suitable model for their specific study.

## 2. In Vitro Models

### 2.1. BC Cell Lines

Cancer cell lines may supply an unlimited source of low-cost and easy-to-handle biological material. For this reason, they are the first key component of basic cancer research studies and drug discovery processes. The first BC cell line was established in 1958 and it is known as BT-20 [12]. Since then, many other BC cell lines have been obtained and characterized, becoming the principal models for BC research. Their naming usually reflects how they are isolated and developed, whether they are obtained from the same patient or the same laboratory. For instance, the most commonly used BC cell line, which is named MCF-7, was obtained at the Michigan Cancer Foundation [13], while the “HCC series” cell lines were established at Hamon Cancer Centre where a successful characterization of a panel of tumor along with paired non-malignant cell lines was achieved [14]. Currently, researchers can choose from hundreds of cancer cell lines having different origins and representing several molecular subtypes. The stratification of this plethora of cellular models based on their molecular characteristics may be a tool to guide the choice of the right model, i.e., the one that is consistent with the objectives of the research one wishes to conduct. One of the most common classifications for BC cell lines is based on the PAM50 (Prediction Analysis for Microarrays) signature [15] conventionally used for BC subtyping. According to the PAM50 classification, they are divided into subgroups showing similar expression levels of the 50-gene signature, namely, luminal, HER2-positive, and basal-like [16,17]. Xiaofeng D. and colleagues classified 84 BC cell lines based on the status of the ER, PR, and HER2 into five subgroups, i.e., luminal A, luminal B, HER2 positive, and basal A, and B, to overcome the inconsistent nomenclature and facilitate BC modeling using appropriate cell lines [18].

#### 2.1.1. Luminal BC Cell Lines

Luminal BC cell lines are characterized by a positive ER and a variable expression of PR. They exhibit high levels of luminal-feature-associated genes, such as GATA3, a critical regulator of luminal-cell differentiation [19], along with epithelial keratins (KRT8, KRT18, and KRT19) [20]. Gene mutation analysis identified a “luminal mutational profile”, which includes mutations in E-cadherin and MAP2K4, as well as amplifications of Cyclin D1, ERBB2, and HDM2. Conversely, the “basal mutational profile” involves BRCA1, RB1, RAS, and BRAF mutations, along with deletions of p16 and p14ARF [21]. Most ER^+^ breast cancers initially respond well to endocrine therapy; however, a significant percentage of these tumors are intrinsically resistant, and another 40% develop resistance to treatment over time, leading to relapse and potential mortality [22]. In this context, the development of endocrine-resistant cell lines has been a global objective for the past few decades. A few ER-positive cell lines (MCF-7, T47D, BT474, and ZR-75) have been extensively utilized in creating various models of acquired anti-hormone resistance in vitro [23], proving to be a valuable tool with high translational potential. For instance, tamoxifen-resistant sublines exhibited a decreased sensitivity to DNA-damaging agents but not to paclitaxel, suggesting that microtubule-targeting drugs may represent the most suitable therapeutic strategy for tamoxifen-resistant BC patients [24]. Collectively, luminal cell lines have undoubtedly advanced our understanding of the estrogen-regulated biology of BC and illuminated the vulnerabilities of ER-positive BC undergoing long-term anti-hormone therapy.

#### 2.1.2. HER2-Positive BC Cell Lines

HER2-positive cell lines are highly heterogeneous, displaying both luminal and basal features [25]. The development of targeted therapies against HER2 has significantly improved the management of HER2-positive BC patients over the years [26]; however, most patients develop resistance to these therapies after a responsive period of variable length. Therefore, enhancing the understanding of the mechanisms that contribute to the onset of resistance is essential for developing therapeutic strategies that may avoid or reverse drug resistance. Generating cell lines resistant to anti-HER2 therapies is an attempt to achieve this goal [27]. Neratinib, an irreversible inhibitor of EGFR, HER2, and HER4 [28], overcame drug resistance in trastuzumab-resistant cell lines, and it showed a synergistic effect when combined with trastuzumab in sensitive cell lines [29,30]. These preclinical findings may provide a rationale for future clinical trials of neratinib in combination with trastuzumab and of neratinib alone in patients refractory to trastuzumab and/or lapatinib treatment.

#### 2.1.3. Basal BC Cell Lines

Basal cell lines are characterized by a low expression or no expression of all three receptors, i.e., ER, PR, and HER2. Gene expression profiling allows us to subdivide them into two distinct groups: basal A cell lines exhibit epithelial-like features and are associated with BRCA1 mutations, while basal B cell lines are characterized by a mesenchymal-like, highly motile, and invasive phenotype, marked by the expression of genes such as CD44 and VIM [25]. The basal B subgroup exhibits a stem-cell-like expression profile and may reflect more closely the clinical “TNBC” tumor type [6,9,25]. Consistently, basal A and basal B cell lines showed a differential sensitivity to a variety of targeted therapies, such as PARP inhibitors, androgen-receptor antagonists, and PI3K signaling inhibitors [31], emphasizing the importance of selecting the appropriate cellular model in drug screening studies.

#### 2.1.4. Characteristics, Strengths, and Weaknesses of Widely Used BC Cell Lines

In Table 1, we report relevant characteristics of the most frequently used BC cell lines. We selected those cell models that were used in more than 10 independent studies in the last 5 years (data retrieved using the PubMed database, September 2024). We highlighted in Table 1 information on the mutational status of TP53, BRCA1, and BRCA2, as they are well-recognized BC susceptibility genes also correlated with patient survival [32]. Finally, since Poly (ADP-ribose) polymerase (PARP) inhibitors have revolutionized the therapeutic landscape of BRCA-associated BC and there is strong evidence that they could also benefit BRCA-non-mutated patients [33], we present the cell line sensitivity to Talazoparib expressed as the half-maximal inhibitory concentration (IC50).

The mutational status of oncogenic genes was extracted from the Cancer Dependency Map (DepMap) portal [34], which offers several key datasets—including the DNA copy number, methylation, mutations, RNAi, protein expression, and drug screening data—of more than 1000 cell lines as part of the Cancer Cell Line Encyclopedia (CCLE). By using genome-wide clustered regularly interspaced short palindromic repeats (CRISPR) and short-hairpin RNA (shRNA) screens, the main aim of the DepMap project is to identify cancer genetic dependencies and chemical sensitivities that may lead to therapeutic vulnerabilities.

Cell lines have been and will remain an irreplaceable tool in preclinical BC research. However, we should ask if and how a continuous cell line may recapitulate the complexity and the heterogeneity of the tumor from which it was initially isolated. It is good news to know that BC cell lines exhibit similar copy number variations (CNVs) and gene mutation patterns as primary tumors, also retaining the same top mutated genes, i.e., TP53 and PIK3CA [16,25].

Single-cell RNA sequencing (scRNA-seq) also demonstrated that cell lines exhibit different drug responses based on their molecular subtype, similar to primary breast tumors [36]. Heiser et al. found that these subtype-associated responses are partly explained by the up- or downregulation of several specific pathways involved in proliferation, cell cycle control, metabolism, adhesion, and invasion [37]. However, the frequency of genetic aberrations is higher in BC cell lines than in early-stage primary tumors, possibly reflecting the fact that many, if not all, of the cell lines commonly used to model BC have been derived from late-stage tumors or pleural effusions [18,38]. Moreover, we should consider that the high level of amplification of survival genes observed in cancer cell lines offers a selective advantage for in vitro growth. In line with this hypothesis, Ben-David U. et al. showed that continuous sub-culturing processes and freezing–thawing cycles led to genetic variations in MCF-7 strains, resulting in differences in proliferation, morphology, and drug responses [39].

An intrinsic limitation of cell lines is that they are usually cultured as monolayers, which do not reflect the complex TME, completely missing the contribution of the extracellular matrix (ECM) and non-malignant cells, including immune and stromal cells. Furthermore, the flattened morphology greatly limits cell-to-cell interactions and results in cells having an equal and constant access to nutrients and oxygen, hampering the development of a transport gradient and making their response to extracellular cues, such as chemotherapy, quite different from the in vivo situation. We found that 3D culture systems hold the potential to completely change the way cancer is modeled in vitro. These models mimic several features of in vivo tumors, including their natural architecture, cell-to-cell and cell–ECM interactions, nutrient and oxygen gradients, drug penetration, and resistance.

### 2.2. Patient-Derived Organoids

Patient-Derived Organoids (PDOs) are 3D models obtained from primary tumors that can reproduce the molecular and histological characteristics of the original tissue. A detailed protocol for developing and propagating in the long term organoids from either normal human breast tissues or BC tissues has been described extensively elsewhere [40]. Sachs et al. created a biobank of 95 BC organoids that represented all the molecular subtypes and conducted high-throughput drug sensitivity screens. Importantly, they observed that the transcriptomic and histological features of the tissue of origin were well-maintained in the organoids, even after 20 passages [41]. Similarly, Chen et al. established PDOs from the tumor tissues of multiple-drug-resistant patients that accurately reflected their previous clinical responses to drug treatments and recapitulated their histopathological and genetic features. Even more importantly, they demonstrated that this PDO platform can be used to predict and guide personalized therapeutic strategies for patients with advanced BC [42]. Notably, scRNA-seq showed that BC PDOs consist of several cell clusters with distinct cellular states and confirmed the existence of high intratumoral heterogeneity in these models, as found in primary tumors [43]. Another application of organoids in BC research is the identification of new immunotherapy strategies as they can recapitulate the complex interaction network between tumor cells and the surrounding immune microenvironment [44]. For instance, mammary ductal epithelial organoids have been used to assess the presence of specific T lymphocyte effector subsets that proliferated in response to an aminobisphosphonate drug and efficiently killed bisphosphonate-pulsed TNBC cells [45].

Given all these applications, organoids hold the potential to become invaluable preclinical tools for translational research. However, their use is not as widespread as that of cancer cell lines due to several limitations. First, their establishment and maintenance are more expensive and have lower success rates than 2D cell culture systems. Moreover, reproducibility is poor, and the features of organoids are not as easy to analyze and measure as those of a cell line [46]. Therefore, culture protocols require further optimization and standardization. Overall, the field of organoid technology for cancer research is still in its early stages, but we believe that the development of organoid biobanks from patients has already contributed and will continue to contribute to a better understanding of BC heterogeneity and the realization of more personalized therapeutic strategies.

### 2.3. Organ-on-Chip Technologies

Traditional organoids are self-organizing 3D structures that can be derived from embryonic stem cells, induced pluripotent stem (iPS) cells, adult stem cells, and patient-derived cancer cells [47]. In contrast, organ-on-chip technologies are microfluidic-based devices made of optically transparent materials, such as polydimethylsiloxane [48]. They represent a more engineered approach to replicating many aspects of the dynamics of an organ in vivo, including biochemical gradients, physical forces, and fluid shear stress [49]. These systems host living cells in a controlled micro- or nano-environment and integrate all the components necessary in order to resemble the tumor vasculature and fluid perfusion that are missing in conventional 3D culture systems [50].

In cancer research, these models are widely used to investigate both the primary and the metastatic TME [51]. Truong et al. developed a 3D organotypic microfluidic platform by co-culturing BC cells with patient-derived CAFs and normal fibroblasts to investigate the role of tumor–stroma interactions in BC progression [52]. They described, for the first time in BC cells, the involvement of glycoprotein nonmetastatic melanoma protein B (GPNMB) in the invasion process promoted by CAFs, demonstrating the potential of this 3D co-culture system to unravel key molecular pathways involved in BC cells–TME interactions. Importantly, several tumor-on-chip platforms have been developed to study BC metastases, each mimicking a different metastatic pattern, such as bone-on-a-chip [53] and lung-on-a-chip [54]. Another relevant application of these technologies is the study of drug pharmacokinetics and pharmacodynamics [55]. A BC-on-chip model was used to test in real time a novel nanoparticle-based doxorubicin delivery system that exhibited higher penetration efficiency and cytotoxicity than free doxorubicin, suggesting that it may be a valuable strategy with which to improve TNBC treatment [56]. More recently, Maulana et al. established a tumor-on-chip platform using a TNBC cell line and primary patient-derived metastatic BC cells that enabled, for the first time, the evaluation of chimeric antigen receptor (CAR)-T cell therapy safety and efficacy, capturing patient-specific responses [57]. The applications described so far highlight that human organ-on-chip technologies can be used to accelerate the discovery of new biomarkers and antineoplastic drugs that hold the promise of an easier clinical translation. However, many challenges remain in developing cancer-on-chip systems before they can be widely applied in industrial and clinical applications. First, their establishment is complicated and requires advanced expertise in nanotechnology and tissue engineering, which can increase experimental costs. Moreover, performance validations and technical reproducibility are not fully achieved. Therefore, until mass standardization is addressed, it is advisable to approach tumor-on-chip models with caution as complementary tools for animal models that still represent the most effective option for preclinical testing.

## 3. Murine Models of BC

Experimental animal models still represent the primary bridge between in vitro studies and clinical trials, as they can better replicate the complex human pathological landscape. Mice are the most used in vivo models in cancer research due to their short reproductive cycles, relative ease and low-cost maintenance compared to other animals, and high degree of gene orthology with humans [58]. Since BC represents a highly heterogeneous and multifactorial disease, several approaches are employed to mimic the human condition in mice, including the use of carcinogen-induced mouse models, germline genetically engineered mouse models (GEMMs), and xenografts.

### 3.1. Carcinogen-Induced Mouse Models

Carcinogen-induced mouse models represent the simplest and most cost-effective animal models used in cancer research. A carcinogen is any biological, physical, or chemical agent that can cause cancer [59]. The carcinogen-induced preclinical models are primarily utilized in the field of cancer prevention, particularly to investigate the effects of various doses of multiple carcinogens and their synergistic or antagonistic impacts, potentially elucidating a threshold level of exposure that may promote tumorigenesis [60]. According to the International Agency for Research on Cancer (IARC) [61], 128 agents are known to be carcinogenic to humans. Among them, 7,12-dimethylbenz[a]anthracene (DMBA), a member of the polycyclic aromatic hydrocarbons (PAHs) classification, can induce BC in mice by causing DNA damage and mutations [62,63]. Abba et al. found that the majority of DMBA-induced breast tumors in mice carry the missense mutation in the kinase domain of PIK3CA, which has been reported as the most frequent PIK3CA mutation in human breast cancers [64]. Therefore, DMBA-induced mouse mammary tumors may serve as an excellent model to closely replicate the biology of PIK3CA-driven mammary carcinogenesis and to evaluate new PIK3CA/AKT/mTOR pathway inhibitory molecules in vivo. However, this system has some limitations since mice can also develop other types of cancer that may affect the interpretation of the results. Additionally, many factors must be defined when planning an experiment using this carcinogen in mice, as the DMBA burden, the number of doses, the exposure duration, and the strain of mice can influence mammary gland tumor incidence and latency, thereby limiting the reproducibility and comparability of the results [65].

### 3.2. GEMMs

The development of GEMMs has been one of the most important milestones in cancer research. In GEMMs, tumors develop de novo in the appropriate tissue, with the appropriate microenvironment, and are capable of recapitulating the genetic background, the main histological subtypes, and the natural biological behavior of human cancers [66]. The development of oncogene-overexpressing transgenic models has been made possible as a result of the identification of mammary-specific promoter elements, including mouse mammary tumor virus (MMTV), long terminal repeat (LTR), whey acidic protein (WAP), C3(1), and bovine β-lactoglobulin (BLG) [67]. Among these, MMTV-LTR is the most commonly used, as it drives transgene expression in the ducts and alveolar cells of the mammary gland at all developmental stages and it is active in both non-lactating and lactating females [68]. MMTV, initially known as the “milk factor” [69], is a retrovirus that can cause mammary tumors in mice by acting as an insertional mutagen or activating the transcription of nearby oncogenes [70]. All existing MMTV-LTR models can exhibit stochastic transgene expression in other secretory tissues, such as salivary glands and seminal vesicles, but in low levels when compared to the mammary glands [68]. In this review, we will limit the discussion to the MMTV-LTR models cited in at least five independent studies in the last 5 years (data retrieved using the PubMed database, September 2024). Table 2 shows the mean tumor latency, the tumor penetrance, and the capacity to develop metastasis of the most frequently used MMTV-LTR mouse models of human BC.

#### 3.2.1. MMTV-PyMT

The MMTV-PyMT (polyomavirus middle T antigen) is the most commonly used GEMM due to the rapid appearance of multifocal mammary adenocarcinomas and lung metastases [71]. This model is easily available from various commercial suppliers around the world. Tumor latency and growth kinetics vary between strains: FVB mice displayed a mean tumor latency of 53 days and exponential tumor growth, whereas C57BL6 mice exhibited delayed tumorigenesis and a more linear growth pattern [72]. The widespread interest in studying PyMT-drive tumorigenesis in mammary epithelium stems from its ability to activate several key signaling pathways involved in human BC progression. Indeed, despite the absence of intrinsic tyrosine kinase activity, this protein interacts with members of the SRC family of tyrosine kinases [77]. Once associated with SRC proteins, PyMT becomes phosphorylated at different tyrosine residues, thereby activating pathways central to malignant transformation, such as the Ras/MAPK cascade and the PI3K signaling (Figure 2) [78,79].

The phosphorylation of different tyrosine residues of the MT protein by SRC kinases leads to the activation of two intracellular pathways that play a significant role in BC growth and progression. The phosphorylation of Y250 mediates the binding to Shc, an adaptor molecule that interacts with Grb2 allowing the recruitment of the Ras exchange factor SOS. Once activated, Ras induces MAPK signaling. The phosphorylation of Y315 enables MT to interact with PI3K, which produces PIP3. This molecule activates PDK1, triggering the Akt/mTOR signaling. (Shc = Src homology and Collagen; Grb2 = Growth factor receptor-bound protein 2; MAPK = Mitogen-activated protein kinase; PI3K = Phosphatidylinositol-4,5-bisphosphate 3-kinase; PIP3 = Phosphatidylinositol (3,4,5)-trisphosphate; PDK1 = 3-phosphoinositide-dependent kinase 1; mTOR = Mammalian target of rapamycin).

A major strength of the MMTV-PyMT model is its faithful recapitulation of the four major stages of human breast carcinogenesis, i.e., hyperplasia, adenoma, and early and late carcinoma. In addition to the morphological similarities, the expression of biomarkers in MMTV-PyMT mice is consistent with those related to poor outcomes in humans, including the loss of ER and PR, as well as the increase in HER2 and cyclin D1 and the loss of integrin-β loss expression when tumors progress to the advanced stages [80]. Moreover, gene expression profiling has revealed a signature associated with high metastatic potential in PyMT tumors, which overlaps with gene signatures predictive of metastasis in human BC [81,82]. Since the key steps of malignant progression from “in situ” lesions to metastasis are so well-mimicked in this model, the MMTV-PyMT has been widely used to investigate virtually all aspects of BC biology. Among the most prominent applications of the model, we can mention the study of molecular signaling driving tumor initiation and progression, the epigenetic modulations in response to environmental stimuli, the role of the host tumor microenvironment in disease progression, and the development and testing of potential therapies, recently extensively reviewed in an excellent manuscript by Attalla et al. [83].

#### 3.2.2. MMTV-NEU

Approximately 15–30% of all breast cancers show HER2/NEU overexpression or amplification [84]. However, definitive evidence that HER2 overexpression alone is sufficient to initiate and drive mammary tumorigenesis is still lacking. To fill this knowledge gap, the Leder laboratory generated the first transgenic mouse model overexpressing HER2 under the control of the MMTV promoter (MMTV-NEU-NT). Mice developed multiple and synchronous mammary tumors with a 3-month latency, suggesting that no additional oncogenic events were required for the transformation of mammary epithelial cells [73]. Later on, Bouchard et al. showed that transgenic mice carrying the activated NEU oncogene under the control of MMTV-LTR developed multiple mammary tumors asynchronously, indicating that NEU overexpression was necessary but not sufficient for the complete malignant transformation of the mammary epithelium [85]. The reasons behind the discrepancies between these models remain unclear but may be attributed to structural differences in the MMTV-NEU fusion transgenes and potential undetected alterations introduced during transgene construction. Over the years, additional MMTV-NEU-NT mice have been generated and commented on extensively elsewhere [86], including an inducible HER2-overexpressing mouse model that employs a tetracycline-regulated system for conditional NEU expression [87]. Following transgene deinduction, all primary and metastatic breast tumors regressed, suggesting that, although several genetic and epigenetic events occur during HER2-induced mammary tumorigenesis, most cells remain dependent upon HER2 for the maintenance of the malignant state. In general, conditional transgenic mice allow the study of the impact of different oncogenes on mammary tumorigenesis, investigating the possible consequences of withdrawing their expression on the maintenance of malignancy and recurrence. Several studies have shown that HER2 may also play a role in regulating the tumor-initiating cell (TIC) population [88], which has proven to be one of the causes of resistance to anticancer therapies and relapses in BC [89]. Therefore, MMTV-NEU models have been used to characterize TICs [90] and to investigate the pathways involved in maintaining HER2-transformed TICs that may have immediate therapeutic implications [91]. Clearly, MMTV-NEU models have been used to test the therapeutic efficacy of many anti-HER2 agents, such as Lapatinib [92], and to study the onset and mechanisms of drug resistance [93,94]. In summary, we can conclude that all current transgenic mouse models of HER2-induced BC have advanced our understanding of how HER2 promotes tumor progression and leads to poor patient prognosis, identifying target genes that function downstream of, or in concert with, HER2. The advantages and limitations of the MMTV-PyMT and MMTV-NEU models are summarized in Table 3.

#### 3.2.3. MMTV-WNT1

The aberrant activation of WNT1 signaling plays a crucial role in the development of several human cancers [95]. The ectopic expression of the *WNT1* gene in MMTV-LTR mice exerts a potent mitogenic effect in epithelial mammary cells, resulting in extensive ductal hyperplasia and the development of early tumors at 5 weeks (WNT1-early) and late tumors at 58 weeks (WNT1-late) of age, which are phenotypically distinct [74]. A hierarchical clustering analysis of the transcriptomic data obtained from WNT1-early and WNT1-late tumors allows for the classification of the MMTV-WNT1 model as “semi-homogeneous” and distinguishes two groups that overlap differently with the human intrinsic subtypes of BC. The analysis also indicates that unique combinations of secondary lesions may cooperate with aberrant WNT1 signaling to target different mammary cell populations, contributing to model heterogeneity [96]. Notably, transcriptomic differences support the phenotypic characteristics of the two groups and their varying sensitivity to Erlotinib, an EGFR inhibitor [97,98].

#### 3.2.4. MMTV-TGFα

The transforming growth factor α (*TGFα*) gene encodes for a 17-kDa protein member of the epidermal growth factor (EGF) family of cytokines. It binds EGFR, activating its downstream signaling pathways that are mainly involved in cell proliferation and migration [99]. To study the role of TGFα in breast tumorigenesis, Matsui et al. developed the first transgenic mouse model overexpressing *TGFα* under the control of MMTV (*MMTV-TGFα*). By 12 months of age, hyperplasia was observed in most females, whereas only 30–40% of them developed adenocarcinomas without any metastases [75,76]. Due to this long tumor latency, the model has been mainly used to study how lifestyle and diet modifications can impact BC development [100,101].

#### 3.2.5. Knockout Models

In addition to oncogene-overexpressing transgenic mice, several tumor suppressor knockout (KO) mouse models have been developed to investigate the impact of their loss of function on mammary tumorigenesis. When the KO is constitutive, the target gene is removed from all tissues simultaneously, often resulting in embryonic or early postnatal lethality. This problem has been addressed by generating conditional KO models, in which gene deletion can be spatially and temporally regulated [102]. One of the most used techniques is the Cre-loxP system, which requires a CRE-driver mouse strain, where Cre recombinase is expressed under the control of cell-specific regulatory elements, and a loxP flanked (floxed) DNA containing strain. Briefly, Cre is a DNA recombinase that specifically recognizes the loxP (locus of x-over, P1) sequence and mediates the site-specific excision of DNA between two loxP sites [103].

As we discussed above, TP53 and BRCA1/2 are high-risk BC susceptibility genes [32]. Therefore, KO mouse models for these genes have been developed. Mice carrying TP53-KO are prone to malignancies in multiple tissues, mainly sarcomas and lymphomas [104]. To overcome the stochastic development of tumors, TP53-floxed mice have been crossed with CRE mice that express the recombinase only in epithelial tissues. For example, Lin et al. crossed conditional TP53^F/F^ mice with MMTV-CRE transgenic strains, creating a model that developed multiple heterogeneous breast tumors, with the tumor latency influenced by the parity of the mice and metastasis to the lungs or liver [105]. Many genomic aberrations of TNBC patients who exhibit simultaneous TP53 downregulation and PI3K-AKT activation are well-mimicked in MMTV-PTEN/TP53 double KO mice and in WAP-CRE, PTEN^F/F^ TP53^R270H^ mutated mice [106,107]. These conditional knockout mice provide suitable preclinical models with which to study basal-like TNBC in an immune-competent environment.

Conditional KO models generated by flanking different exons with loxP sites allowed us to overcome the embryonic lethality associated with the constitutive homozygous deletion of BRCA1 [108]. For instance, the WAP- or MMTV-CRE-mediated excision of exon 11 caused mammary-gland tumor formation after a long latency, suggesting that BRCA1 inactivation does not directly promote tumorigenesis, but it may increase genetic instability. Moreover, a significant acceleration in tumor formation was observed after introducing one TP53-null allele, providing evidence that p53 is involved in BRCA1-related tumorigenesis [109]. Notably, Hollern et al. demonstrated that mice carrying the simultaneous tissue-specific deletion of TP53 and BRCA1 (K14-CRE; TP53^F/F^ BRCA1^F/F^ mice) give rise to mammary neoplasms with genomic and transcriptomic similarities to human basal-like BC [110]. Similar results regarding tumor latency and synergistic activity with p53 have emerged in conditional WAP- and K14-CRE BRCA2-knockout mice [111,112]. Nevertheless, these models developed more heterogeneous tumors with a much higher incidence and larger sizes, suggesting that BRCA1 and BRCA2, despite some similarities, may promote BC development in quite different ways. This is consistent with the fact that human BRCA1- and BRCA2-associated BCs differ in several aspects, including gene expression profiling [113] and gender-specific risk [114].

The generation of the clustered regularly interspaced short palindromic repeats (CRISPR)-associated protein 9 (CRISPR/Cas9) system has improved genome editing in mice, reducing the time to develop both knockouts and knockins. Two components are required to perform genome editing: the Cas9 nuclease, which creates double-stranded DNA breaks (DSBs), and a “guide RNA”, which directs Cas9 to a specific target DNA site [115]. The co-injection of Cas9 nuclease mRNA with one or more “guide RNA” molecules directly into fertilized eggs enables the one-step generation of single- or multiple-gene mutant mice [116]. Pioneering work from Annunziato et al. demonstrated that the somatic engineering of the mammary gland by the intraductal injection of lentivirally encoded “guide RNAs” in female Cas9 knock-in mice is a feasible and effective strategy with which to model invasive lobular breast carcinoma and TNBC [117,118]. However, a major concern in the applications of the CRISPR/Cas9 system is the off-target effects, although several in silico methods have been developed in the last decade to predict them [119]. Another issue that still requires improvement involves the delivery methods to enhance specificity and efficiency while minimizing the host’s immune response.

#### 3.2.6. Current Challenges and Future Perspectives

Although GEMMs have revolutionized preclinical studies in human BC research by mimicking many pathophysiological and molecular features, their implementation in experimental routines remains limited by the significant time and resources required to generate and maintain them. Furthermore, GEMMs do not fully recapitulate the biology of human tumors. For example, metastatic spread occurs only through hematogenous dissemination to the lungs and lymph nodes in MMTV-PyMT and MMTV-NEU mice [71,73], while each subtype of human BC is associated with distinct patterns of metastatic spread [120]. Importantly, species differences often make the extrapolation of results from GEMMs to humans unreliable. Recently, the anti-HER2 antibody–drug conjugate demonstrated high anti-tumor activity and was well-tolerated in MMTV-HER2 models but caused several persistent toxicities in patients with HER2^+^-metastatic BC, resulting in the early termination of the phase I clinical trial [121]. An emerging approach to enhancing the translational power of GEMMs in clinical settings is integrating omics technologies as complementary tools that analyze their genomic, transcriptomic, and proteomic landscapes. The whole-genome sequencing of the MMTV-PyMT and MMTV-NEU models identified several similarities and differences compared to the human disease from the perspectives of SNV, CNV, and translocation, demonstrating that a comprehensive genomic characterization of GEMMs must be considered when they are selected as model systems for BC [122]. Similarly, Pfefferle et al. employed a transcriptomic approach to profile tumors from 27 GEMMs, illustrating how these mice recapitulate human BC subtypes and providing valuable insights for accurately designing a preclinical study [96]. More recently, the application of scRNA-seq enabled the creation of a single-cell atlas of epithelial and immune cells from all four tumor progression stages of the MMTV-PyMT mouse model, leading to the identification of a highly heterogeneous and dynamic stem-like cell cluster [123]. ScRNA-seq has also been utilized to profile cancer-associated fibroblasts (CAFs) isolated from the MMTV-PyMT model, revealing three transcriptionally and functionally distinct subpopulations that correlate with BC biological behavior and prognosis [124]. All these studies emphasize how integrating omics datasets may guide the selection of an appropriate GEMM model for a specific BC subtype and enhance the translation of preclinical findings into clinically relevant information.

### 3.3. Xenograft Models

Xenograft models allow us to overcome some of the important limitations discussed for GEMMs, including expensive and time-consuming experimental protocols and the non-human nature of the tumor mass. Cell-line-derived xenografts (CDXs) and patient-derived xenografts (PDXs) are commonly used translational models for the preclinical evaluation of the therapeutic efficacy and toxicity of new compounds before moving to clinical trials.

#### 3.3.1. CDX Models

CDXs are based on the engraftment of human cell lines to immunodeficient mice to avoid the graft-versus-host reaction of the mouse against the human tumor tissue. These models include orthotopic and ectopic xenografts, originating either from the implantation of human cancer cells in the same organ from which the cancer originated in humans or from their systemic delivery (subcutaneous, intraperitoneal, intravenous, or intramuscular) [125]. Since metastatic BC is one of the main causes of BC-related deaths, investigating the mechanisms by which tumor cells spread and select specific organs to invade is one of the main applications of CDX models. Each human breast tumor subtype has been correlated with a specific pattern of metastatic spread, revealing notable differences in patients’ survival. High rates of bone metastases are found in luminal and HER2^+^ subgroups, while the brain, liver, and lungs are the predominant sites of metastasis for the TNBC subtype [120,126]. Unfortunately, ER^+^ BC cell lines used for the generation of CDXs (MCF-7, T47D, and ZR75 cells) produce poorly invasive tumors that seldom metastasize [127]. Instead, most CDX metastasis models rely on TNBC cell lines exhibiting markers of aggressiveness and invasion [128]. One of the most commonly used cell lines is MDA-MB-231, whose transcriptomic profile partially overlaps with that belonging to basal-like metastatic BC samples [129]. Lateral tail vein injections of MDA-MB-231 primarily result in lung metastases [130], while intracardiac injections promote dissemination to the bone and brain [131,132]. The ability of MDA-MB-231 cells to efficiently metastasize to bone is associated with the expression profile of a defined set of genes that closely match the gene profiling of human bone metastases [133]. More generally, the metastatic behavior of a cell line in vivo depends on the site of inoculation and the intrinsic tropism of the tumor cells. Jin, X. et al. developed a high-throughput method for the simultaneous in vivo testing (in a single mouse) of the metastatic potential of hundreds of cancer cell lines. By employing an in vivo barcoding strategy, they investigated the metastatic potential and the organotropism of 21 basal-like BC cell lines in the CCLE and created a first-generation metastasis map (MetMap) that reveals organ-specific patterns of metastasis, allowing these patterns to be associated with clinical and genomic features [133]. In addition to MDA-MB-231, other BC cell lines, such as T-47-D [134] and SUM-149 [135], have been used to establish metastatic models through ectopic administration. These systems show a rapid development of metastases, but they do not describe the early stages of the metastatic cascade since tumor cells are directly introduced into the vascular system [136]. This issue can be at least partially addressed by developing orthotopic models in which BC cells are engrafted into the mammary fat pads of mice, giving the possibility to resemble all steps of the metastatic dissemination.

Although CDXs have proven to be very useful in studying the biology and metastatic potential of BC, they suffer from intrinsic limitations. Firstly, before engraftment, cell lines are cultured in vitro for many passages, being subjected to genetic drift and clonal selection. Then, once injected into the host mice, they grow as a quite homogenous mass of malignant breast epithelial cells that cannot recapitulate the heterogeneity of human breast tumors and their microenvironment. Secondly, the cell lines used are mainly derived from late-stage breast tumors or pleural effusions, making them not functional for modeling early events of BC progression and for studying pathways involved in the acquisition of endocrine therapy resistance [25].

#### 3.3.2. PDX Models

To address the shortcomings discussed above, PDXs obtained following the transplantation of primary human cancer cells or small tumor pieces into host mice have been developed. Since these models are established directly from patient tissues, they maintain the 3D architecture and reflect the heterogeneity of human tumors, providing an intermediate complexity model that may help to bridge the experimental data obtained in vitro with those obtained in a real clinical setting. Unlike CDXs that can be generated in nude mice, PDX generation requires severe immunodeficient mice, including non-obese diabetic (NOD)-scid, NOD.Cg-PrkdcscidIl2rgtm1Wjl/SzJ (NSG), or NOD.Cg-PrkdcscidIl2rgtm1Sug/Jic (NOG) mice [137].

Although many academic and commercial sources of PDXs are available, the methods for establishing and characterizing these models are poorly standardized and yield inconsistent published results. To circumvent this obstacle and assist investigators in finding relevant data for their own research, the major PDX consortia, the NCI-PDXNet Portal [138] and the EuroPDX [139], have defined the PDX models’ Minimal Information standard (PDX-MI) for quality assurance and the use of PDX models. This resource describes the minimal clinical information required to define the patient’s tumor as well as the information about the process of generating and validating the corresponding PDX model [140]. Several studies have demonstrated that PDXs display the same morphological and functional features as the original tumors. For instance, DeRose et al. showed that breast tumor grafts maintained the histopathological characteristics, sites of metastasis, clinical markers, estrogen dependence for ER^+^ neoplasms, and gene expression profiles of the parental tumors [141]. Although hundreds of PDXs have been developed for BC [142], representing the deadliest subtypes is still challenging. Recently, Guillen et al. developed a biobank of PDXs and matched organoids from tumors representing the greatest unmet needs, including endocrine-resistant, treatment-refractory, and metastatic breast tumors [143]. They demonstrated that these models are valuable systems for drug development and functional precision medicine in real time with clinical care. As a proof of concept, orthotopic PDXs of HER2^+^ and HER2-low BC brain metastases have proven to be robust models for evaluating the efficacy of trastuzumab deruxtecan on tumor growth and overall survival [144]. Moreover, since metabolic dysregulation is one of the main features of BC [145], Liao et al. performed metabolic profiling and a gene expression analysis, comparing TNBC and ER^+^ patient samples, as well as TNBC PDXs, to identify new therapeutic vulnerabilities. They found two major metabolic clusters independent of BC histological subtypes that are associated with different patient prognoses [146,147]. All their findings are made available for data mining through a freely accessible portal to assist future investigations [148].

Despite all the advantages of PDX model systems, there are also some limitations. First, the overall take rates of transplanted tumors largely differ based on tumor types of origin and are, in general, very low [149]. Moreover, these models are established in immunodeficient mice; therefore, they cannot be used to study the critical role played by the immune system in tumor progression or to develop novel immune therapies. In recent years, to satisfy this unmet need, humanized mouse models, which consist of immunodeficient mice co-engrafted with human tumors and immune cells, have been generated.

#### 3.3.3. Humanized Mouse Models

Immunotherapy has emerged as a promising treatment for advanced BC and may revolutionize the clinical management of hard-to-treat BC patients. However, the limited number of preclinical models that accurately recapitulate interactions between the human immune system and tumors hampers the research progress in this field. An emerging approach is the creation of humanized mouse models that consist of immunodeficient mice engrafted with functional human immune system cells. Thus far, three types of humanized mice have been established: hu-PBL (peripheral blood lymphocytes), hu-CD34^+^, and BLT mice (bone marrow–liver–thymus) [150]. In the BC field, hu-CD34^+^ models have been used to establish TNBC PDXs for the preclinical investigation of immunotherapeutic approaches, such as anti-PD1-based therapies [151]. Moreover, the same strategy facilitated the development of the first PDX model of ER^+^ ESR1-mutant endocrine-resistant BC, for which new therapeutic options are urgently needed [152]. Hu-CD34^+^ mice can be used as platforms for the validation of novel combinations of chemotherapy and immunotherapy, as demonstrated for the first time by Burlion et al., who showed that the combination of cyclophosphamide with an immunotherapeutic antibody was more effective in reducing tumor growth than single treatments [153]. More recently, the combination of anti-PD1 antibodies and anti-angiogenic agents as a new promising therapeutic strategy for TNBC patients was investigated in hu-PBL mice, revealing that this neoadjuvant therapy may synergistically prevent postoperative tumor recurrence and metastasis [154].

Despite the significant potential of these models to recapitulate a functional human immune system and their emerging role in precision medicine, there are challenges that scientists worldwide must address. First, graft-versus-host disease (GvHD) is a common limitation in humanized mouse models, resulting in short observable and therapeutic window periods [155]. Second, ethical concerns arise from the use of fetal tissues, such as the liver and thymus, thus hampering large-scale mouse production for drug testing. In addition, the most commonly used mouse strains, such as NSG and NOG, do not support the development of the human myeloid compartment and negatively affect natural killer cell expansion and motility due to the insufficient cross-talk between the murine cytokines and the human cells [156]. The supplementation of the corresponding human cytokines by administering recombinant proteins [157] or using immunodeficient transgenic mice expressing the relevant proteins [158] has improved the humanization of the model, but investigators must be aware of the potential negative consequences of the persistent presence of these inflammatory cytokines in mice [159]. Finally, the protocols adopted by laboratories can be substantially different, with a negative impact on model reproducibility. An attempt to reduce this high variability has been the creation of an international resource called “Minimal Information for Standardization of Humanized Mouse Models” (MISHUM), whose aim is to promote protocol exchange and transparency in reporting the analyses performed as well as the results obtained in order to improve the usability and credibility of each model [160].

## 4. Ethical and Financial Issues in Choosing Animal Models

One of the main aspects researchers must consider before choosing the most suitable animal model for their experimental purposes is the ethical aspect, as animal welfare is becoming increasingly valued. In Europe, the use of animals for research purposes is regulated by the European Convention for the Protection of Vertebrate Animals Used for Experimental and Other Scientific Purposes [161], which was adopted by the Directive 2013/63/EU of the European Parliament and the Council on the Protection of Animals Used for Scientific Purposes. These guidelines aim to ensure the application of the 3R Principles, which stand for Replacement, Reduction, and Refinement [162]. First proposed in 1959, they encourage avoiding the use of experimental animal models by employing alternative valuable systems or, if it is not possible, the use of the smallest number of animals required to achieve the scientific goal. Moreover, they emphasize the importance of limiting animal physical distress and pain in order to avoid unpredictable biases that may compromise the scientific outcomes of the research. Besides these ethical considerations, financial and logistical issues also greatly affect the choice of the best animal model. The development of Sharing Experimental Animal Resources: Coordinating Holdings—Breast (SEARCHBreast), a database where researchers can find and share materials related to animal models of BC, helps researchers to save time and money, complying, at the same time, with the 3R Principles [163]. Finally, animal models often fail to translate drug findings from bench to bedside since humans and mice do not respond the same to the same treatment due to species-specific characteristics. Indeed, the highest percentage of failed phase II and phase III clinical trials in the oncology field is mainly due to a lack of either efficacy or safety [164]. Failure rates can be reduced by implementing more stringent criteria during the preclinical stages, including target validation as well as lead compound discovery and optimization, and carefully choosing the most suitable animal model. The evaluation of more clinically relevant endpoints, such as progression-free survival instead of tumor volume reduction and the number of distant metastases, along with more accurate preclinical testing in fit-for-purpose in vivo models, might help bridge the translational gap between preclinical and clinical results. A summary of the main advantages and limitations of all BC murine models is presented in Table 4.

## 5. Conclusions

An experimental model, by definition, is not a perfect replication of the clinical condition; rather, it helps to model, in a simplified setting, specific aspects of human pathology. The choice of the most suitable model must be tailored to each project, considering the research topic, the scientific question, the level of evidence the study aims to reach, logistic issues, and, last but not least, ethical concerns. In Figure 3, we schematically highlight the suitability of different in vitro and in vivo models of BC based on the specific research field. A combinatorial experimental design that combines different models may help to overcome the inherent limitations of single systems.

We found that 2D cultured cell lines represent the most convenient way to study the molecular details of intracellular and intercellular mechanisms under controlled and reliable environmental conditions. However, 3D in vitro models, as PDOs and organ-on-chip technologies, better mimic the heterogeneity of BC tumors and hold the promise of advancing individualized BC treatments. We are confident that the significant technological advancements in the field will contribute to improving the reliability of 3D platforms, which hold the potential to change the paradigm of preclinical research. However, mouse models remain an essential reference and a mandatory step for BC translational studies as they continue to provide valuable insights into complex whole-body interactions.

In our opinion, as omics technologies continue to evolve and generate more comprehensive datasets, they will become even more critical in understanding the translational relevance of established and emerging BC models to bridge the gap between basic science and clinical research.

## Figures and Tables

**Figure 1 ijms-26-04572-f001:**
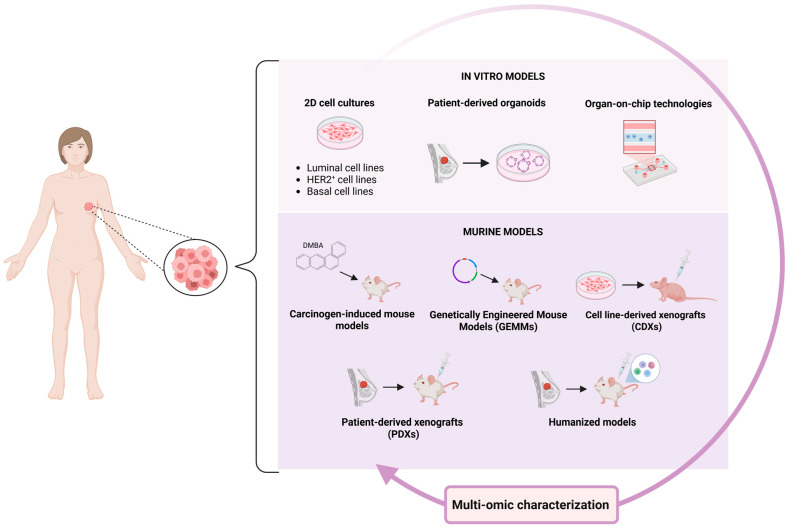
Preclinical models for BC. Schematic representation of widely used BC preclinical models including both in vitro and in vivo systems.

**Figure 2 ijms-26-04572-f002:**
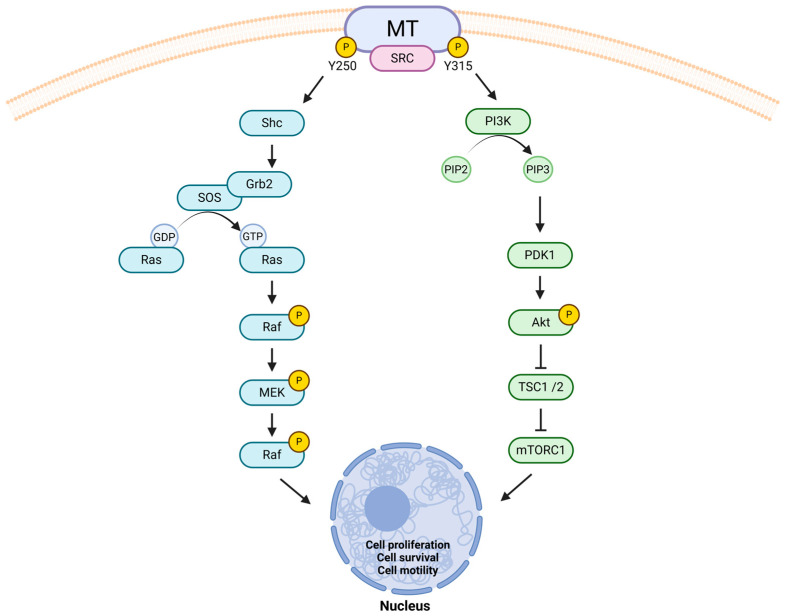
Signaling pathways downstream of MT phosphorylation involved in breast tumorigenesis.

**Figure 3 ijms-26-04572-f003:**
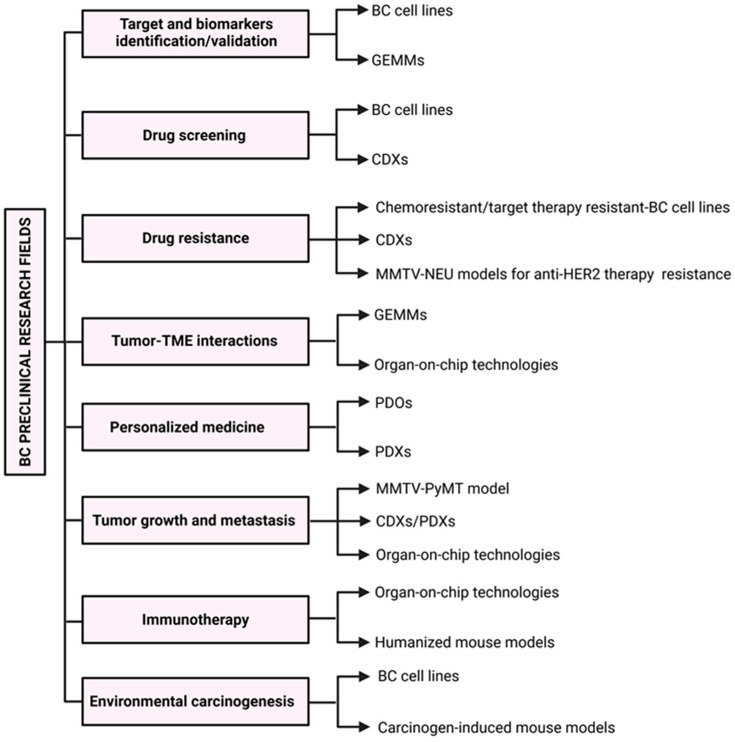
Main applications of BC preclinical models. Schematic representation of the main applications of preclinical models in the most important BC research fields to guide the selection of the most suitable model for each specific experimental question.

**Table 1 ijms-26-04572-t001:** Summary of relevant features of the most used BC cell lines.

Cell line	ER ^#^	PR ^#^	HER2 ^#^	TP53 Status ^$^	BRCA1 Status ^$^	BRCA2 Status ^$^	Additional Oncogenic Mutated Genes ^$^	Talazoparib IC50 (μM) *
Luminal cell lines								
BT474	+	+	+	SNV	WT	SNV	*PIK3CA*, *RIT1*, *RHOA*	670.8
CAMA-1	+	+/−	-	SNV	WT	WT	*PTEN*	61.2
MCF-7	+	+	-	WT	WT	WT	*PIK3CA*	77.5
MDA-MB-361	+	+/−	+	SNV	WT	SNV	*PIK3CA*	182.2
T-47-D	+	+	-	SNV	WT	WT	*PIK3CA*	70.6
ZR-75-1	+	+/−	-	WT	WT	WT	*PTEN*	NA
ZR-75-30	+	-	+	WT	WT	SNV	/	463.1
HER2-positive cell lines								
AU565	-	-	+	SNV	WT	WT	/	16.4
HCC1954	-	-	+	SNV	WT	WT	*PIK3CA*, *GAB1*	87.3
MDA-MB-453	-	-	+	WT	WT	WT	*PIK3CA*, *FGFR4*	186.5
SKBR-3	-	-	+	SNV	WT	WT	/	NA
Basal A cell lines								
BT20	-	-	-	SNV	WT	WT	*PIK3CA*	127.9
HCC1143	-	-	-	SNV	WT	WT	*FGFR2*	44.6
HCC1806	-	-	-	insertion	WT	WT	/	NA
HCC1937	-	-	-	SNV	insertion	WT	/	118.9
HCC70	-	-	-	SNV	WT	WT	/	32.6
MDA-MB-436	-	-	-	insertion	SNV	WT	/	49.2
MDA-MB-468	-	-	-	SNV	WT	SNV	/	27
Basal B cell lines								
BT549	-	-	-	SNV	WT	WT	*PTPRT*	42
CAL-51	-	-	-	WT	WT	WT	*RRAS2*, *PIK3CA*	0.8
HCC38	-	-	-	SNV	WT	WT	/	39.8
HCC1395	-	-	-	SNV	SNV	SNV	/	11.9
HS578T	-	-	-	SNV	WT	WT	*HRAS*	97.3
MDA-MB-157	-	-	-	deletion	WT	WT	*RAC1*	122.9
MDA-MB-231	-	-	-	SNV	WT	WT	*KRAS*	35.5
SUM-149-PT	-	-	-	SNV	deletion	WT	/	NA
SUM-159-PT	-	-	-	insertion	WT	WT	*HRAS*, *PIK3CA*	NA

^#^ Receptor status and molecular subtypes were assigned according to Dai et al. [18]. ^$^ Mutational status was extracted from the DepMap portal [34] * The IC50 concentration data were extracted from Genomics of Drug Sensitivity in Cancer (GDSC) [35]. SNV = single nucleotide variant; WT = wild-type.

**Table 2 ijms-26-04572-t002:** Most common MMTV-LTR models.

Transgene	Mean Tumor Latency (Days)	Tumor Penetrance	Metastatic	Reference	Citations on PubMed *
*PyMT*	53–92	100%	Yes	[71,72]	165
*Neu*	90	100%	Yes	[73]	28
*Wnt-1*	35–406	80%	No	[74]	16
*TGF* *α*	480	30–40%	No	[75,76]	7

* Number of independent studies published in the last 5 years retrieved through a search on the PubMed database (last access September 2024).

**Table 3 ijms-26-04572-t003:** Advantages and limitations of MMTV-PyMT and MMTV-NEU mouse models.

GEMMs	Advantages	Limitations
MMTV-*PyMT*	Rapid appearance of multifocal mammary adenocarcinomas Tumor progression recapitulates the four main stages of human breast carcinogenesis Biomarkers expression consistency	Tumor latency and growth kinetics variable between strains No distinct patterns of metastatic spread
MMTV-*NEU*	Describe HER2-related tumorigenesis Allow for testing of anti-HER2 therapies	Inconsistencies between models of different laboratories No distinct patterns of metastatic spread

**Table 4 ijms-26-04572-t004:** Comparison of BC murine models.

**Murine Model**	**Advantages**	**Limitations**
Carcinogen-induced mouse models	Low-cost Easy establishment	Off-target effects Variable tumor incidence and latency
CDXs	Rapid establishment Allow for in vivo drug testing Can be used to study metastasis to different organs	BC cell lines subjected to genetic drift and clonal selection Tumor heterogeneity not recapitulated Missing early events of BC progression and metastasis
PDXs	Genetically and histologically similar to human BCs Can mirror tumor heterogeneity Allow for testing personalized therapies Biobanks	Limited number of patient samples Low transplant rates Immunodeficient mice
Humanized models	Genetically and histologically similar to human BCs Can mirror tumor heterogeneity Allow for testing immunotherapy	Low transplant rates GvHD Limited reproducibility

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
