# Peer review of "Facing the Challenge to Mimic Breast Cancer Heterogeneity: Established and Emerging Experimental Preclinical Models Integrated with Omics Technologies"

_ijms, 2025, doi:10.3390/ijms26104572_

Round 1

Reviewer 1 Report

Comments and Suggestions for Authors

The manuscript provides a comprehensive and well-organized review of preclinical models used in breast cancer research, with a particular focus on the application of omics technologies. Overall, the review is clear, informative, and well-structured. No major criticisms are noted.

Minor suggestions include expanding the discussion in section 3.1.5 to cover additional conditional knockout (CKO) mouse models beyond MMTV-Cre, particularly WAP-Cre models, which are more commonly used in studies of inherited breast cancers such as Brca1, Brca2, and Palb2 CKO models.

Additionally, there are a few typos throughout the manuscript that should be carefully checked and corrected.

Author Response

The manuscript provides a comprehensive and well-organized review of preclinical models used in breast
cancer research, with a particular focus on the application of omics technologies. Overall, the review is
clear, informative, and well-structured. No major criticisms are noted.
Q1: Minor suggestions include expanding the discussion in section 3.1.5 to cover additional conditional
knockout (CKO) mouse models beyond MMTV-Cre, particularly WAP-Cre models, which are more
commonly used in studies of inherited breast cancers such as Brca1, Brca2, and Palb2 CKO models.
A1: We thank Reviewer 1 for this important suggestion. In the revised manuscript, we clearly specified the
type of promoter used for the conditional expression of Cre. In addition to MMTV, some studies already
included in the first version of the manuscript, used WAP and K14. We also added one more study (ref
number 107) that used WAP-CRE conditional knockout mice to study the effect of TP53 mutation in PTEN
deleted mice. All changes are highlighted in yellow.
Q2: Additionally, there are a few typos throughout the manuscript that should be carefully checked and
corrected.
A2: We carefully checked and corrected for typos throughout the manuscript.

Reviewer 2 Report

Comments and Suggestions for Authors

This is an important review that can help researchers understand and choose the best models to perform critical tests to study BC. I only have a few recommendations.

How are 3D organoids and Organ-on-chip performed? It is important to highlight the differences between them.

The authors did not address another important preclinical model of BC: Chemically induced tumors in mice. Please, include it. This review would also benefit from the description of the other GEMMs, or at least explain further (an overview) why the authors did not discuss them.

Round 2

Reviewer 2 Report

Comments and Suggestions for Authors

The authors answered my comments appropriately, though they could have made a better discussion about chemically induced tumor models and how they could mimic the TME of certain tumors.